# Integrated in Silico and Experimental Approach towards the Design of a Novel Recombinant Protein Containing an Anti-HER2 scFv

**DOI:** 10.3390/ijms22073547

**Published:** 2021-03-29

**Authors:** Joana Santos, Miguel Cardoso, Irina S. Moreira, João Gonçalves, João D. G. Correia, Sandra Cabo Verde, Rita Melo

**Affiliations:** 1Centro de Ciências e Tecnologias Nucleares, Instituto Superior Técnico, Universidade de Lisboa, Estrada Nacional 10, ao km 139, 7, 1649-004 Lisbon, Portugal; joana.d.santos@tecnico.ulisboa.pt (J.S.); jgalamba@ctn.tecnico.ulisboa.pt (J.D.G.C.); sandracv@ctn.tecnico.ulisboa.pt (S.C.V.); 2Faculty of Pharmacy, iMed.ULisboa—Research Institute for Medicines, University of Lisbon, 1649-004 Lisbon, Portugal; mgcardoso@ff.ulisboa.pt (M.C.); jgoncalv@ff.ulisboa.pt (J.G.); 3Department of Life Sciences, University of Coimbra, Calcada Martim de Freitas, 3000-456 Coimbra, Portugal; irina.moreira@cnc.uc.pt; 4Center for Neuroscience and Cell Biology, Center for Inovative Biomedicine and Biotechnology, 3000-456 Coimbra, Portugal; 5Departamento de Engenharia e Ciências Nucleares, Instituto Superior Técnico, Universidade de Lisboa, Estrada Nacional 10, ao km 139, 7, 1649-004 Lisbon, Portugal

**Keywords:** cell transfection, DNA plasmid, human epidermal growth factor receptor 2, molecular docking, recombinant protein

## Abstract

Biological therapies, such as recombinant proteins, are nowadays amongst the most promising approaches towards precision medicine. One of the most innovative methodologies currently available aimed at improving the production yield of recombinant proteins with minimization of costs relies on the combination of in silico studies to predict and deepen the understanding of the modified proteins with an experimental approach. The work described herein aims at the design and production of a biomimetic vector containing the single-chain variable domain fragment (scFv) of an anti-HER2 antibody fragment as a targeting motif fused with HIV gp41. Molecular modeling and docking studies were performed to develop the recombinant protein sequence. Subsequently, the DNA plasmid was produced and HEK-293T cells were transfected to evaluate the designed vector. The obtained results demonstrated that the plasmid construction is robust and can be expressed in the selected cell line. The multidisciplinary integrated in silico and experimental strategy adopted for the construction of a recombinant protein which can be used in HER2+-targeted therapy paves the way towards the production of other therapeutic proteins in a more cost-effective way.

## 1. Introduction

The human epidermal growth factor receptor 2 (HER2) is a transmembrane tyrosine kinase receptor which has an important function in cell growth, survival and differentiation [1]. As it plays a key role in cell survival, its level of expression affects cell behavior, being overexpressed in 30% of breast cancers [2]. Moreover, it is associated with tumor aggressiveness and a higher probability of relapse. The most widely used drug for targeting HER2 is trastuzumab, a humanized monoclonal antibody that recognizes an extracellular domain of HER2, ultimately blocking HER2 effects and decreasing cancer cell growth and survival [1]. The latest advances in recombinant antibody technology have greatly facilitated the genetic manipulation of antibody fragments [3]. Thus, it has become possible to develop a wide variety of artificial antibody molecules for research, diagnosis, and therapy. Single-chain variable domain fragments (scFv) are amongst the several antibody fragments used in research and clinical settings, mostly due to better pharmacokinetic properties when compared to a whole antibody, namely, better tissue penetration and rapid blood clearance [4].

One way to enhance the therapeutic efficacy of a drug is to increase specificity towards its putative target, falling into the category of the so called targeted therapies. Additionally, nanoparticles opened up new perspectives for the next generation of targeted therapies due to their recognized ability to improve drug packaging, delivery and targeting efficiency [5]. Among others, the viral nanoparticles (VNP) appear as promising and exciting nanoplatforms to be used as drug delivery systems due their biocompatibility and biodegradability [6].

To acquire specificity, VNPs need to express a cell target-specific molecule of interest, such as a recombinant protein on the surface. For this, the important advances in biomedical research through the rapid process of recombinant DNA production and protein engineering have been crucial to the success of these new approaches [7]. For efficient and selective targeting, the cell-recognizing protein needs to have a high functional affinity, low dissociation rate, appropriate biodistribution characteristics and low immunogenicity [8]. Plasmids are very commonly used as vectors to express the desired protein. Several types of plasmids are commercially available for cloning and gene expression in a wide variety of hosts, such as *Escherichia coli*, yeast and mammalian cells [9].

With the development of state-of-the-art sequencing technologies, the size of biological databases has increased dramatically, becoming easy to obtain biological sequences in a quick and inexpensive way. However, it is relatively slow and expensive to extract functional and structural information due to the limitations of conventional biochemical assays [10]. In order to overcome wet experimental costs, a data-driven docking strategy can be efficiently used to evaluate protein structure and function analysis [10].

Taken all together, our main aim was the design and production of a biomimetic vector containing the scFv of trastuzumab, a well-known anti-HER2 antibody, as a targeting motif fused with HIV viral protein gp41. HIV-based VNPs show strong in vivo immunogenicity [11] and have been amplified in clinical environment [12]. Docking studies were performed in order to select the most favorable residue in the viral protein to be considered to fuse with a scFv from trastuzumab, after which DNA plasmid was produced. Here, we intended to provide a novel approach to constructing a new recombinant protein from two separate ones. This undocumented approach can provide a new strategy for targeted therapy using recombinant proteins to be expressed into VNPs that will be designed by in silico methods.

## 2. Results

We begin by describing the results attained from the computational study which aimed at the prediction of the best model of a protein-based sequence containing an anti-HER2 scFv and viral protein gp41 by molecular docking, followed by the construction of the plasmid and its validation through a transfection protocol and Western blot analysis. This combined approach proved to be a useful tool that can be applied to other systems.

### 2.1. Docking Simulation

To develop a computational model of the selected scFv from trastuzumab and viral protein gp41, we performed molecular docking using the HADDOCK (High-Ambiguity-Driven protein–protein DOCKing) web server [13]. HADDOCK has been used previously to predict protein–protein interactions [14], and molecular docking analyses have been successful in similar studies [15,16]. Among the 200 docking decoys obtained from HADDOCK, we selected the top 10 through the discrete optimized protein energy (DOPE) score and the MODELLER objective function. Next, the most favorable site to be linked to the scFv was based on the minimum distance between the N-terminal of scFv and the membrane-proximal external region (MPER) of gp41 (Figure 1).

Our study identified ASP12 as the most promising residue with less conformational and energetic disorder (Figure 1A, inset). In addition to the minimal distance between Cα of both proteins, the MPER of gp41 was considered in order to keep the transmembrane region to allow its expression onto a VNP surface. Thus, the fact that the TM of the viral protein was not affected assured that in the transcription process of the recombinant protein, the function of gp41 was not affected. A new model was constructed (Figure 1B) linking the N-terminal of scFv from trastuzumab and ASP12 Cα of gp41. This new sequence-based structure was minimized, and the conformation was not affected.

Having now the complete sequence of the new protein, the next step was the construction and transformation of the plasmid.

### 2.2. Vector Design and Assessment of the Recombinant Protein Expression

The vector (pcDNA3.1+) used for mammalian expression contains protein sequence scFv-HER2_gp41 (inserted into cloning site NheI/NotI) and, for the use in DNA cloning, a pUC replication origin and an ampicillin resistance gene (AmpR) for selection and maintenance in *E. coli*, a neomycin–kanamycin resistance gene (NeoR/KanR) for selection of stable cell lines, a human cytomegalovirus (CMV) promoter/enhancer for constitutive expression of the protein of interest and two tags for protein expression detection, N-terminal HA-tag and C-terminal c-Myc (Figure 2). The expected molecular weight of the recombinant protein is 47.1 kDa.

In order to evaluate the ability of the constructed plasmid to be expressed inside cells, HEK-293T cells were transiently transfected with the X1665 vector. After 48 h post-transfection, the cells were collected and lysed to assess through the Western blot technique the expression of the plasmid in this cell line. Figure 3 shows the immunoblot analysis with an anti-HA antibody and the presence of scFv-HER2_gp41 expressed intracellularly in HEK-293T cells.

The molecular weight of the protein expressed by the cells is about 49 kDa (Figure 3, lane 3), slightly higher than the theoretical value. These results confirm the presence of the scFv-HER2_gp41 protein in the cell extract.

## 3. Materials and Methods

### 3.1. Molecular Modeling and Docking Simulation

The 3D structures of scFv and gp41 were constructed by homology modeling using the MODELLER package [19]. This software allows the construction of 3D models for a protein with unknown structure by using one or more proteins with structural information as templates. The scFv from trastuzumab was constructed based on PDB ID 3AUVA [20]. A multi-template modeling protocol was used for gp41 model construction according to the different regions of the viral protein. Thus, the membrane proximal external region (MPER) of HIV-1 gp41 (PDB ID 2X7RC [17]) was chosen for the MPER region and the transmembrane region was constructed based on the HIV-1 gp41 transmembrane domain (PDB ID 2MG1A [18]). The alignments between the target and the selected templates were combined into a multiple alignment using the target sequence as an anchor [21]. The best models from each query were selected using the discrete optimized protein energy (DOPE) score [22] and the MODELLER’s objective function [23].

For the docking simulation studies. we used the HADDOCK (High-Ambiguity-Driven protein–protein DOCKing) (http://haddock.science.uu.nl/services/HADDOCK/haddock.php, accessed on 13 January 2020) server [13]. HADDOCK starts with a randomization of orientations and rigid body energy minimization (1000 solutions) followed by semi-rigid simulated annealing in a torsion angle space (200 solutions) and final refinement in a Cartesian space with an explicit solvent (200 solutions). It uses biological information to drive the docking by introducing AIRs (ambiguous interaction restrains). The gp41 active residues chosen were the ones in the MPER (GLU7-LYS30) that were close enough to the transmembrane region (LYS30-ILE51). The runs were performed using different subsets of these residues, in particular, PHE20-TRP25, ALA1-ILE29 and ARG8-LEU16. Among the 200 protein–peptide complexes obtained for each run, the distance between the C-terminal of each active residue and the N-terminal from the first amino acid in the scFv sequence were measured using an in-house R script [24].

### 3.2. Plasmid DNA Construction and Transformation of Escherichia coli

The plasmid DNA containing the scFv-HER2_gp41 protein sequence was synthesized in Synbio Tech (South Brunswick, NJ, USA) and replicated in *Escherichia coli*. The transformation of *E. coli* was performed by electroporation, following a similar protocol reported by Lessard [25].

### 3.3. Cell Culture and Transfection

HEK-293T cells (human embryonic kidney-273 cells expressing the large T antigen of simian virus 40) were cultured in a Dulbecco’s modified Eagle’s medium (DMEM) supplemented with 6 mM L-glutamine, 1 mM sodium pyruvate, 0.1 mM MEM Non-Essential Amino Acids, 10% (*v*/*v*) of heat-inactivated fetal bovine serum and 1% (*v*/*v*) penicillin–streptomycin, all obtained from Cytiva (Marlborough, MA, USA). The cells were cultured in 75 cm^3^ disposable polycarbonate flasks at 37 °C with 5% of CO_2_. The cells were seeded at 0.5 × 10^6^ cell/well in a six-well plate and transfected using Lipofectamine 3000 (Lipo3; Cat# L3000001; Thermo Fisher Scientific, Waltham, MA, USA) at 3.75 µL/well and with 2.5 µg/well of DNA according to the manufacturer’s instructions. The transfected cell cultures were maintained at 37 °C with 5% CO_2_. At 24 h post-transfection, the medium was replaced with a new pre-warmed medium supplemented with kanamycin (500 µg/mL). At 48 h post-transfection, the cells were washed with the phosphate-buffered saline (PBS) and lysed with 150 µL of the lysis buffer (CellLytic^TM^M Cell; Sigma Life Science, St. Louis, MO, USA). Protease inhibitors (Roche, Basel, Switzerland) were added to the previous lysis mixture according to the manufacturer’s instructions and incubated at 4 °C for 15 min with gentle agitation. The cell extract was then centrifuged at 4 °C for 15 min at 20,000 g.

### 3.4. Western Blot

Total protein concentration of the supernatant was determined using the DC Protein Assay (BioRad, Hercules, CA, USA) according to the manufacturer’s instructions. Expressed proteins in the cell extracts were visualized by Western blotting. A total of 25 μg of protein was mixed with 1/6 of a 6X sample buffer and boiled for 10 min at 95 °C. The samples were loaded per lane and migrated to a 12.5% SDS-PAGE gel. After electrophoresis, the samples were transferred onto a nitrocellulose membrane (BioRad). The membrane was blocked at room temperature for 60 min with 5% (w/v) non-fat dried milk in PBS containing 0.2% (*v*/*v*) Tween 20 (PBSTween). After it was washed with PBSTween, the membrane was incubated for 90 min at room temperature with gentle agitation with an HRP (horseradish peroxidase)-conjugated anti-HA polyclonal antibody (Fisher Scientific) diluted 1:2500 in 1% (w/v) non-fat dried milk in the PBSTween. Then, it was washed with PBSTween and the proteins were visualized using the ECL^®^ reagent (GE Lifesciences, Chicago, IL, USA) according to the manufacturer’s instructions. The molecular weight observed in the Western blot of the scFv-HER2_gp41 protein was obtained using the ImageJ program [26].

## 4. Conclusions

Molecular modeling and docking studies were performed to design the recombinant protein sequence containing the scFv of an anti-HER2 antibody fragment as a targeting motif fused with HIV gp41. Based on those results, a robust DNA plasmid was constructed and expressed in the HEK-293T cell line. The aim of this communication was achieved as the recombinant protein was successfully constructed. Nevertheless, its functionality will have to be evaluated, as well as the ability of the newly designed vector to be expressed into an HIV1-based VNP. Our work simultaneously contributed to the development of a methodology that will ultimately lead to the improvement of the production yield of recombinant proteins while minimizing costs by relying on the incorporation of the power of computational tools to foresee the predicted recombinant protein before experimental validation.

This is a ground-breaking approach to improve the design and preparation of target-specific recombinant proteins for the development of biological therapies towards precision medicine.

## Figures and Tables

**Figure 1 ijms-22-03547-f001:**
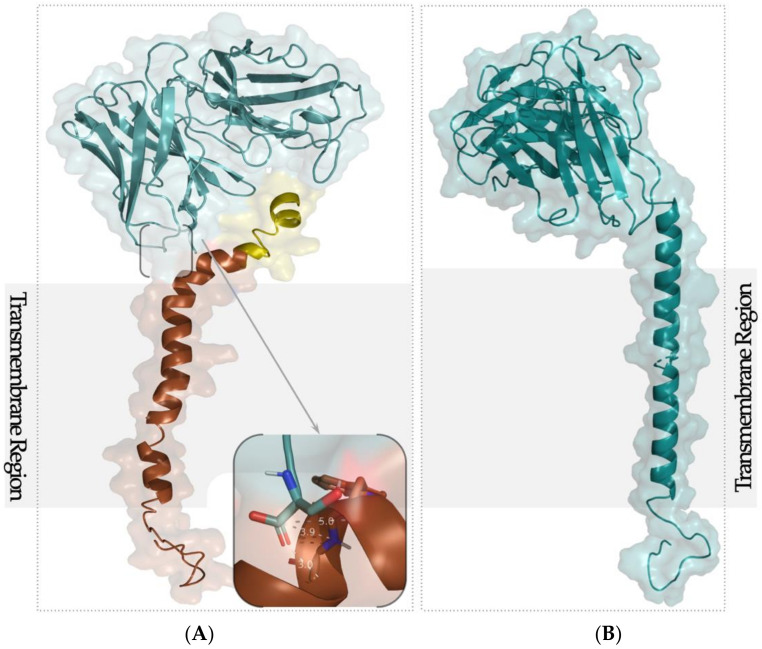
(**A**) 3D model of gp41 (PDB ID 2X7RC [17] and 2MF1A [18], brown) and scFv from trastuzumab (PDB ID 3AUVA [16], blue) (inset: distance between Cα of the C-terminal of scFv and ASP12 of the MPER of gp41); (**B**) new sequence-based protein structure model. The transmembrane region (TM) is identified by the grey region.

**Figure 2 ijms-22-03547-f002:**
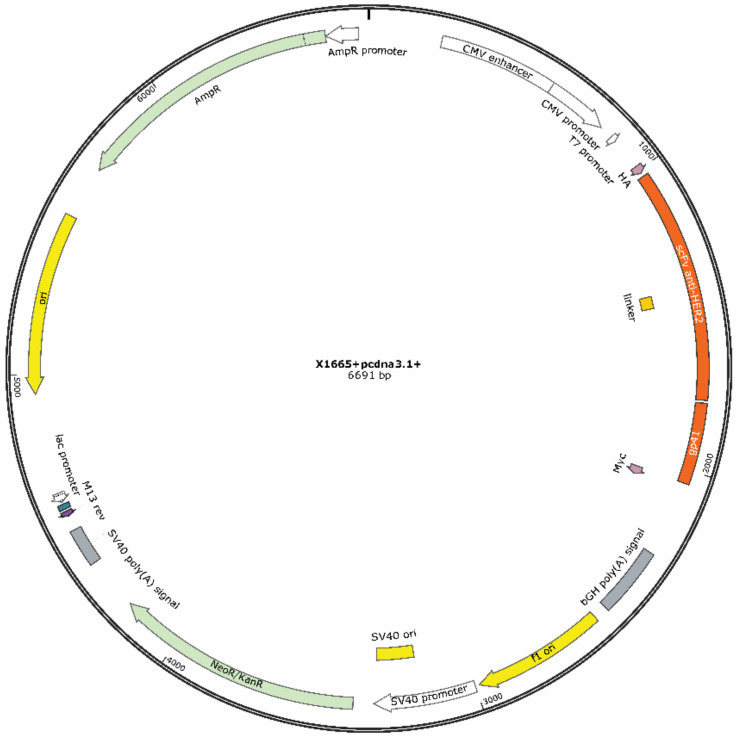
Constructed DNA plasmid encoding the scFv-HER2_gp41 sequence.

**Figure 3 ijms-22-03547-f003:**
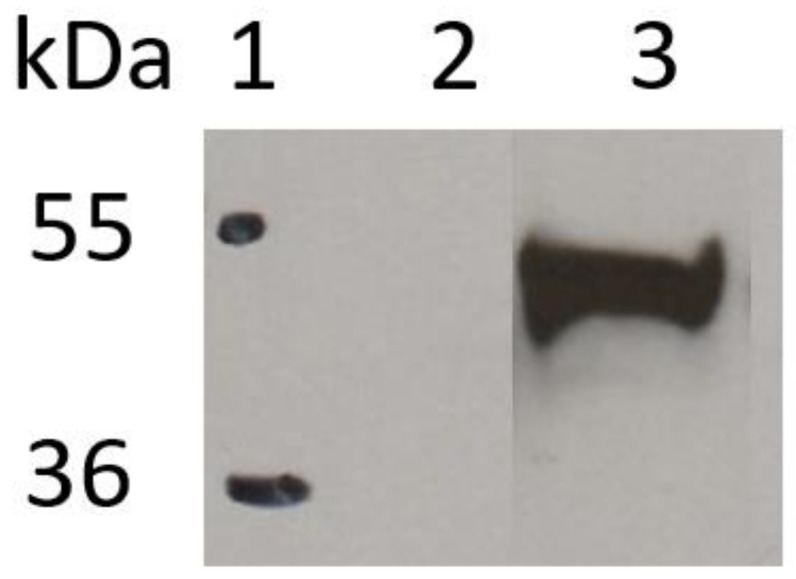
Western blot analysis. Lane 1: protein ladder, lane 2: negative control (transfection without a plasmid), lane 3: scFv-HER2_gp41 protein (transfection with plasmid X1665). Western blot detection was performed using an HRP-conjugated anti-HA polyclonal antibody (1:2500). Visualization of bands was carried out using the ECL^®^ reagent.

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
