# Peer review of "Integrated in Silico and Experimental Approach towards the Design of a Novel Recombinant Protein Containing an Anti-HER2 scFv"

_ijms, 2021, doi:10.3390/ijms22073547_

Round 1
Reviewer 1 Report
Santos et al. ijms-1125113 " Integrated in silico and experimental approach towards the design of a novel recombinant protein containing an anti-HER2 scFv" is a valuable review paper that shows the importance of combining in silico studies as a strategy for recombinant protein production. The authors have produced a novel recombinant protein with antiHER2scFv and gb41. The strategy employed in constructing this recombinant protein is interesting as it could be instrumental in making other therapeutic proteins. However, the reviewers suggest adding results to increase this paper's value and hope that providing more information (described below) will improve this study's quality.
As shown in Figure 3, the authors have confirmed the presence of scFv-HER2_gp41 protein in the cell extracts. However, the authors did not show its characterization or biological activity. The reviewers believe that the authors need to investigate the stability of the scFv-HER2_gp41 protein in solution and whether it binds to HER2.
Reviewer 2 Report
In this manuscript the authors computational method to study the protein-protein interaction between the scFv of Trastuzumab and HIV Gp41. The results are used to design a new chimeric containing both proteins. Next, a mammalian expression vector was produced containing the DNA of the new construct, transfected into HEK293T cells and the expression checked by Western Blot.
The experimental part of this work is very limited. The authors only checked the expression of the chimeric protein by Western. This only shows that a protein containing an HA-tag of the expected size is expressed. It says nothing about the quality and functionality for the produced protein. Hereto, the protein should have been purified and characterized by several biophysical methods such MS, DLS, etc. This would also have given the authors the possibility to compare the actual protein with the model proposed in Figure 1B.
Other comments:
- In the legend of Figure 1 the authors mention that the transmembrane domain™ is identified by the grey region. However, this is not visible in the figure.
- There is no Chapter 3.
Round 2
Reviewer 1 Report
The authors answered the reviewer’s comments and revised the manuscripts appropriately. Although unknown problems remain for future work, the reviewer considers the paper accepted for publication in the journal.
Reviewer 2 Report
In the revised manuscript not much has changed. It is still my opinion that to be able to claim 'experimental approach' (as in the title) more experimental work should have been done.
There is still no Chapter 3.